# Increased Glycated Hemoglobin but Decreased Cholesterol after a Loss of *Helicobacter pylori* Infection: A Community-Based Longitudinal Metabolic Parameters Follow-Up Study

**DOI:** 10.3390/jpm11100997

**Published:** 2021-09-30

**Authors:** Li-Wei Chen, Cheng-Hung Chien, Chih-Lang Lin, Rong-Nan Chien

**Affiliations:** 1Department of Gastroenterology and Hepatology, Chang-Gung Memorial Hospital and University, Keelung Branch, Keelung 204, Taiwan; leiwei@cgmh.org.tw (L.-W.C.); cashhung@cgmh.org.tw (C.-H.C.); lion@cgmh.org.tw (C.-L.L.); 2Community Medicine Research Center, Chang-Gung Memorial Hospital and University, Keelung Branch, Keelung 204, Taiwan

**Keywords:** *Helicobacter pylori*, metabolic syndrome, glycated hemoglobin, body mass index, HDL

## Abstract

This study aimed to evaluate the impact of *Helicobacter pylori* (*H. pylori*) infection on metabolic parameters in a longitudinal follow-up manner. From August 2013 to August 2019, a community-based prospective study of *H. pylori* and metabolic syndrome (MetS) was performed in the northeastern region of Taiwan. A total of 1865 subjects were divided into four groups according to the serial results of urea breath test (UBT): new *H. pylori* infection (group 1, *n* = 41), null *H. pylori* infection (group 2, *n* = 897), loss of *H. pylori* infection (group 3, *n* = 369), and persistent *H. pylori* infection (group 4, *n* = 558). When comparing the subjects between groups 1 and 2, HBA1c was associated with a new *H. pylori* infection. Body mass index (BMI) was associated with a loss of *H. pylori* when comparing subjects between groups 3 and 4. Elevated HBA1c and high-density lipoprotein (HDL) levels but lower values of cholesterol and white blood cells (WBCs) were found during serial analyses within group 3. Conclusively, HBA1c was associated with a new *H. pylori* infection. BMI was associated with *H. pylori* loss. Increased HBA1c and HDL values but decreased values of cholesterol and WBC were associated with a loss of *H. pylori* infection.

## 1. Introduction

*Helicobacter pylori* (*H. pylori*) infection is the most common chronic bacterial infection in humans. Serologic evidence of *H**. p**ylori* infection is uncommon in children before age 10 but rises to 10% in adults between 18 and 30 years of age, and further increases to 50% in those age 60 or older [1,2]. *H. pylori* infection induces local inflammation in the stomach and a systemic immune reaction in the whole body [3]. The actions of the virulence factors of *H. pylori* include flagella, urease, lipopolysaccharides, adhesins to stimulate Lewis x antigen, type IV secretion of CagA and exotoxin of Vac A, lytic enzymes, and heat shock proteins [3]. Some immune related cells, such as mast cell, macrophage, and T-cell, are involved in *H. pylori* infected reaction. Mast cells were abundant in the mucosa of antral gastritis and had a positive correlated with polymorphonuclear and mononuclear cell infiltration [4,5]. *H. pylori* infection is associated with increased expression of the macrophage migratory inhibitory factor (MIF) protein and MIF mRNA in gastric epithelial and inflammatory cells; along with other cytokines, MIF may play a significant role in gastric inflammation [6]. T cell (e.g., Th1 cell) activation by *H. pylori* infection may contribute to more severe inflammatory condition [3]. Elevated levels of inflammatory cytokines, such as interleukin 1, 8, 17, tumor necrosis factor α (TNF-α), and lowered levels of leptin are involved in these inflammatory reactions [3,7,8]. A leptin deficiency and high levels of inflammatory cytokines are the critical mechanisms for insulin resistance (IR) and metabolic syndrome (MetS) [8,9,10,11]. *H. pylori* infection-related extragastric diseases, such as IR, dyslipidemia, obesity, and MetS have been reported [3,12,13,14]. Our previous studies revealed subjects aged less than 50 y/o with *H. pylori* infection increased a risk of being obesity (BMI ≥ 30) (Odd ratio, OR = 1.836), IR (OR = 12.683), and MetS (OR = 3.717) compared to those without *H. pylori* infection [2,15].

The hypothesis in this study was that a new infection or a loss of *H. pylori* infection might change the local gastric and systemic immune reaction, which would influence the metabolic parameters, such as sugar, lipids, and weight during the follow-up period. The individuals without a change in the *H. pylori* infection status served as the control group. This prospective, community-based study aimed to evaluate the differences in the metabolic parameters between subjects with or without *H. pylori* infection (intergroup analysis) and the change in metabolic parameters within a new infection or a loss of infection of *H. pylori* (intragroup analysis).

## 2. Materials and Methods

This study was a community-based survey for *H. pylori* and MetS in adults, performed in the northeastern region of Taiwan from August 2013 to August 2019. Participants from the four districts of Anle (metropolis), Gongliao (rural area), Ruifang (mountain city), and Wanli (fishing village) were included. The inclusion criteria were individuals 30 years or older because MetS is mostly diagnosed in middle-aged people or older (>30 years old) and due to ethical considerations [16]. The exclusion criteria were pregnancy or lack of adequate follow-up data, including lesser than four follow-up urea breath tests (UBT) and blood tests. All subjects participated in a demographic survey, physical examination, blood tests, and underwent a UBT for *H. pylori* infection survey annually. The demographic survey assessed the history of systemic or malignant diseases, such as hypertension (HTN), diabetes mellitus (DM), hyperlipidemia, hematologic disorders, medication history, and family history. Medications, such as proton pump inhibitors or histamine 2 blocker agents, were suspended at least for 14 days before UBT. Physical examination included heart rate, blood pressure, body weight, body height, and waist girth (circumference). The body mass index (BMI, kg/m^2^) was calculated as the weight (kg) divided by squared height (m). The subjects were asked to fast overnight before drawing the blood samples. Blood tests included a complete blood cell count; metabolic parameters, including fasting sugar, total cholesterol, triglyceride, high density lipoprotein (HDL), glycated hemoglobin (HbA1C), insulin levels, and C-reactive protein (CRP) levels. The blood samples were analyzed within 4 h after collection. The Institutional Review Board of the Chang-Gung Memorial Hospital approved this research (IRB No: 103-3886C). All participants agreed to participate in the study and signed the informed consent form before enrollment.

### 2.1. Urea Breath Test

The C^13^-UBT was performed after an overnight fast using the Proto Pylori kit (Isodiagnostika, Canada) containing 75 mg of C^13^-urea and additives. The subjects were defined as new *H. pylori* infection when the serial UBT results were negative, negative, positive, and positive (new infection group). Likewise, the subjects were defined as “loss of *H. pylori* infection” when the serial UBT results were positive, positive, negative, and negative (loss of infection group). The subjects who showed consistent results of UBT, such as “positive, positive, positive, positive” or “negative, negative, negative, negative,” were defined as “persistent infection group” or “no infection group.” Because there was no other test except UBT to confirm the status of *H. pylori* infection in this study, subjects with only one positive or only one negative result, inconsistent results, such as positive, negative, negative, positive or negative, positive, positive, negative, in the serial four UBT were excluded.

### 2.2. Metabolic Syndrome

A race-specific waist girth threshold based on the National Cholesterol Education Program Adult Treatment Panel (NCEP ATP) III criteria were used to prevent distortions in the MetS prevalence [17]. The cut-off values for a normal waist circumference in Asian men and women were set to 90 cm (35.4 inches) and 80 cm (31.5 inches), respectively.

### 2.3. Risk Factors of H. pylori Infection

*H. pylori* infection’s risk factors include age, socioeconomic status, urban-dwelling (different districts), number of family members, and the level of education [18].

### 2.4. Data Analysis and Statistics

For continuous variables, the values are expressed as mean ± standard deviation (SD). A t-test was applied to compare the mean values of two independent samples. A paired sample t-test was applied for a repeated measures analysis within the same sample. A one-way ANOVA was used to compare the mean values of multiple samples. The categorical data were analyzed using the Chi-squared test or Fisher’s exact test, as appropriate. A case-control matched analysis by propensity score was used in this observational study to reduce the selection bias. All the statistical tests were two-tailed. A *p*-value of <0.05 was considered to be statistically significant. A logistic regression analysis was applied for the odds ratio (OR) of a new onset of *H. pylori* infection when comparing the subjects with a new *H. pylori* infection to subjects without an *H. pylori* infection. Likewise, the OR of a loss of *H. pylori* infection was analyzed when comparing subjects with a new onset of *H. pylori* infection in subjects with a persistent *H. pylori* infection. First univariate regression analysis was performed for candidate factors, then multivariate regression analysis was performed for selective factors. A repeated test from stored serum would be performed for the management of missing data.

Statistical analyses were performed using PASW for Windows (version 18.0, SPSS Inc., Chicago, IL, USA).

## 3. Results

The patients were divided into four groups: new *H. pylori* infection group (group 1, *n* = 41), null *H. pylori* infection group (group 2, *n* = 897), loss of *H. pylori* infection group (group 3, *n* = 369), and persistent *H. pylori* infection group (group 4, *n* = 558) (Figure 1). Table 1 shows the demographics of these four groups. When comparing the subjects between groups 1 and 2, older age (>60 years), living in Wanli district, and having more family members (6–10) were found in group 1 (new *H. pylori* infection) (*p* < 0.05).

There was no difference in the age, gender, districts¸ education level distributions, and number of family members between the subjects of groups 3 and 4. Among 369 subjects in group 3 (loss of *H. pylori* infection), 186 (50.4%) subjects had received *H. pylori* eradication therapy, and 143 (38.8%) subjects had received endoscopic studies at our hospital.

When comparing the subjects with and without new *H. pylori* infections (group 1 vs. group 2), baseline HBA1c was observed to be associated with new *H. pylori* infections (adjust odds ratio, aOR = 1.338, 95% CI: 1.034—1.731, *p* < 0.027, adjusting the confounding factors of age and gender) (Table 2).

When comparing the subjects with and without loss of *H. pylori* infection (group 3 vs. group 4), the baseline body weight (aOR = 1.019, 95% CI: 1.004–1.034, *p* < 0.013) and BMI (aOR = 1.068, 95% CI: 1.027–1.109, *p* < 0.001) were associated with the loss of *H. pylori* infection. Table 3 shows the intragroup parameter changes in Groups 1–4 by the paired *t*-test (Table 3).

In group 3 (loss of *H. pylori* infection), elevated HbA1C and HDL values were found (mean delta change of HBA1c = 0.123, *p* < 0.014; mean delta change of HDL = 2.028, *p* < 0.001). However, lowered values of WBC and cholesterol were detected (mean delta change of WBC = −0.281, *p* < 0.001; mean delta change of Cholesterol = −6.723, *p* < 0.001) (Table 3). Among groups 1, 2, and 4, no significant differences in the values were found. In group 4 (persistent *H. pylori* infection), a more increased HBA1c (mean delta change = 0.327, *p* < 0.003) was found when compared subjects in group 3.

For the common diseases including DM, HTN, dyslipidaemia, CKD, and MetS, no significant difference in the disease status (new onset or a loss of disease) was found between those with or without a change in the *H. pylori* infection within the four years of follow-up (Table 4).

## 4. Discussion

Most individuals experience *H. pylori* infection during childhood, and the infection rate increases with age [1]. This increased prevalence of infection with age was initially thought to represent a continuing acquisition throughout adulthood. However, new infection or reinfection in adults is quite uncommon, especially in developed countries, where it is estimated to occur in less than 0.5% of cases per year [19]. Demographic factors, such as old age, number of family members, and district differences were related to a new *H. pylori* infection in the present study, when compared to subjects without new *H. pylori* infections.

In the current study, glycated hemoglobin (HBA1c) is observed to be associated with new *H. pylori* infections among subjects without prior *H. pylori* infections. HBA1c results from hemoglobin glycosylation, reflecting integrated blood glucose levels during the preceding three months [20]. Previous studies revealed an association between a post-*H. pylori* infection status and insulin resistance or glycated hemoglobin [12,20,21]. However, there are no reports regarding the relationship between serum HBA1c levels and new *H. pylori* infection. The present study demonstrated that HBA1c is associated with a new *H. pylori* infection (OR = 1.338, 95% CI: 1.034–1.731, *p* < 0.027). A possible explanation is that a high blood glucose level results in immune dysfunction and gastrointestinal dysmotility, which increases the vulnerability to an infection in the stomach [22]. In one study, young patients with diabetes presented a higher risk of *H. pylori* reinfection following previous successful *H. pylori* eradication [23].

The current study revealed that weight and BMI were associated with a loss of *H. pylori* infection. This observation may be not a causality result. In this study, we found subjects with high body weight and BMI had more hospital visits for underlying diseases. During outpatient clinics, they might have additional chances for gastrointestinal referrals and receiving *H. pylori* eradication therapy.

Boyuk et al. reported no associations between *H. pylori* presence and inflammatory response, which was evaluated by neutrophil/lymphocyte ratio (NLR) and platelet/lymphocyte ratio (PLR) measurements in patients with dyspepsia [24]. They though *H. pylori* infection might not be related to the chronic inflammatory response. But our prospective study revealed a decreased WBC in follow up data among those loss of *H. pylori* infection. A further prospective study is required to clarify the change of NLR or PLR and the relationship between these inflammatory changes and gastric malignant formation.

Although several studies have reported the associations between IR, MetS, CKD, or dyslipidemia, and *H. pylori* infections, these extra-gastric diseases are not recommended for routine checkups of *H. pylori* infection and treatment according to current guidelines [3,25,26]. Our study further demonstrated that a change in the *H. pylori* infection status was not associated with new onset or a loss of some common diseases, such as DM, HTN, dyslipidemia, CKD, and MetS. The influence of *H. pylori* eradication on glucose or lipid control has been controversial in previous studies [27,28,29,30]. The current study revealed that a loss of *H. pylori* infection would increase the value of HDL and HBA1c but decrease the values of cholesterol and WBC. Some studies reported the impact of gut microbiota change on the metabolic parameter following *H. pylori* eradiation therapy [31,32]. These changes of metabolic parameters may have only minor influence on the progression or resolution of diseases, such as DM, HTN, or CKD within four years of follow-up.

Because of the increasing antibiotic resistance of ***H. pylori***, the eradication rate of clarithromycin-based triple therapy has decreased below 80% [3,33]. Other combinations for eradication therapy, such as concomitant therapy, hybrid and high dose dual therapy have all been used as first-line therapies [34,35]. Some gastrointestinal symptoms, such as reflux symptoms, maldigestion or abdominal pain, and hematologic diseases, such as iron deficiency or thrombocytopenia purpura, may be relieved after eradication therapy [36]. To overcome the varying potency of PPIs interfered by CYP2C19 genotypes, several studies used a potassium-competitive acid blocker, P-CAB, such as vonoprazan for ***H. pylori*** eradication combination therapy [37].

This study has a few limitations. First, a selection bias may be present. Our study originated from a community-based survey for MetS. Although the data collected from the community subjects might be nearer the real-world condition, the participants in our study were older (mean age > 50 years) and female predominant (more than 60%). The participants may consist of older adults with underlying diseases and a desire for medical examinations. Second, several subjects did not complete the four UBT follow-ups. Some participants declined a follow-up UBT, especially when a negative result of UBT was found in previous studies. The other participants had inconsistent UBT results (unsure *H. pylori* infection status) were excluded for analyses. Third, a few subjects who took antibiotics from other hospitals might recall as no antibiotic usage on questionnaire screening. The number of cases of spontaneous loss of *H. pylori* infection might be overestimated.

## 5. Conclusions

HBA1c was associated with a new *H. pylori* infection (OR = 1.338). Weight and BMI were associated with a loss of *H. pylori* infection (OR = 1.019 and 1.068, respectively). By intra-group analysis, increased values of HBA1c and HDL but decreased values of cholesterol and WBC were detected among subjects with a loss of *H. pylori* infection. However, the progression of diseases such as DM, HTN, CKD, dyslipidemia, and MetS were not different after the changes in *H. pylori* infection status within four years of follow-up.

## Figures and Tables

**Figure 1 jpm-11-00997-f001:**
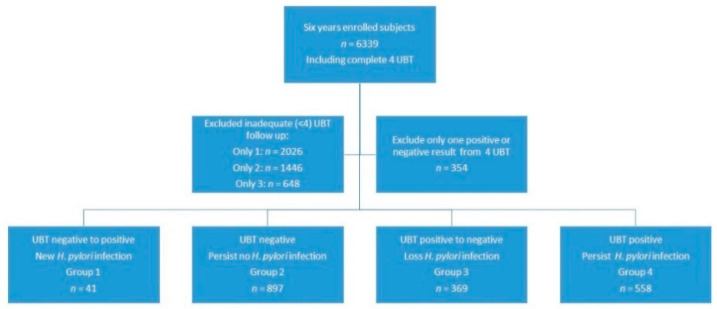
Study diagram.

**Table 1 jpm-11-00997-t001:** Demography.

	Group 1	Group 2	Group 3	Group 4	*p* Value
Number	41	897	369	558	
Age	62.1 ± 10.4	54.3 ± 12.2	57.8 ± 10.3	59.1 ± 11.1	<0.001
Gender (%)					<0.006
Male	13 (31.7)	256 (28.5)	112 (30.4)	208 (37.3)	
Female	28 (68.3)	641 (71.5)	257 (69.6)	350 (62.7)	
Age Group (%)					<0.001
30–39	1 (2.4)	145 (16.2)	24 (6.5)	27 (4.8)	
40–49	3 (7.3)	147 (16.4)	43 (11.7)	78 (14.0)	
50–59	11 (26.8)	277 (30.9)	134 (36.3)	173 (31.0)	
60–69	14 (34.1)	238 (26.5)	126 (34.1)	183 (32.8)	
70+	12 (29.3)	90 (10.0)	42 (11.4)	97 (17.4)	
District (%)					<0.001
Anle	25 (61.0)	608 (67.8)	215 (58.3)	307 (55.0)	
Gongliao	2 (4.9)	78 (8.7)	45 (12.2)	52 (9.3)	
Ruifang	2 (4.9)	94 (10.5)	62 (16.8)	107 (19.2)	
Wanli	12 (29.2)	117 (13.0)	47 (12.7)	92 (16.5)	
Education Level (%)					<0.001
Illiterate	5 (12.2)	46 (5.1)	30 (8.2)	55 (9.8)	
Primary school	11 (26.8)	194 (21.6)	102 (27.6)	146 (26.2)	
Junior high school	8 (19.5)	138 (15.4)	60 (16.3)	105 (18.8)	
Senior high school	14 (34.1)	264 (29.4)	121 (32.8)	154 (27.6)	
College	3 (7.3)	224 (25.0)	54 (14.6)	89 (15.9)	
Graduate school	0 (0.0)	31 (3.5)	2 (0.5)	9 (1.6)	
Family member (%)					<0.025
1–5	34 (82.9)	858 (95.7)	350 (94.9)	525 (94.1)	
6–10	7 (17.1)	34 (3.8)	19 (5.1)	30 (5.4)	
11–15	0	5 (0.5)	0	3 (0.5)	

Group 1: a new onset of getting *H. pylori* infection; Group 2: no *H. pylori* infection; Group 3: a new onset of losing *H. pylori* infection; Group 4: a persist *H. pylori* infection.

**Table 2 jpm-11-00997-t002:** Factors associated with *H. pylori* infection status change by logistical regression analysis.

A New *H pylori* Infection (Group 1 vs. Group 2)						
Univariate			Multivariate			
Parameter	aOR	95% CI	*p*-Value	aOR	95% CI	*p*-Value
waist	1.013	(0.983, 1.045)	0.403			
BMI	1.039	(0.956, 1.128)	0.372	1.016	(0.905, 1.037)	0.463
FBG	1.006	(0.993, 1.020)	0.339			
HBA1c	1.338	(1.034, 1.731)	<0.027	1.470	(1.045, 1.851)	<0.021
Triglyceride	1.001	(0.997, 1.006)	0.537			
Cholesterol	0.998	(0.990, 1.005)	0.518			
HS CRP	0.936	(0.776, 1.129)	0.487			
WBC	1.124	(0.906, 1.394)	0.289			
A loss *H pylori* infection (Group 3 vs. Group 4)						
	Univariate			Multivariate		
Parameter	aOR	95% CI	*p*-Value	aOR	95% CI	*p*-Value
waist	1.011	(0.995, 1.027)	0.196			
BMI	1.068	(1.027, 1.109)	<0.001	1.274	(1.154, 1.376)	<0.001
FBG	1.004	(0.999, 1.009)	0.131			
HBA1c	0.908	(0.797, 1.036)	0.151	1.049	(0.845, 1.161)	0.233
Triglyceride	1.000	(0.998, 1.002)	0.911			
Cholesterol	0.999	(0.995, 1.003)	0.600			
HS CRP	0.977	(0.929, 1.027)	0.355			
WBC	0.948	(0.861, 1.043)	0.272			

aOR: adjusted odds ratio, after adjusting factors of age and gender; FBG: fasting blood glucose; WBC: white blood cell; HS CRP: high sensitive C-reactive protein; HBA1C: glycated hemoglobin; BMI: body mass index.

**Table 3 jpm-11-00997-t003:** Intragroup mean values change after a different infection status of *H. pylori*.

A New *H. pylori* Infection (Group 1)				A Loss of *H. pylori* Infection (Group 3)		
Parameter	Mean ^†^	T	*p*-Value	Mean	T	*p*-Value
weight	−0.131	−0.546	0.586	−0.100	−0.805	0.422
waist	−1.031	−1.904	0.060	0.447	1.733	0.084
FBG	−3.531	−1.723	0.089	1.169	0.792	0.429
HBA1c	0.011	0.050	0.960	0.123	2.477	<0.014
Triglyceride	−4.383	−0.688	0.493	0.757	0.224	0.823
Cholesterol	−1.321	−0.425	0.672	−6.723	−3.746	<0.001
HDL	1.510	1.210	0.230	2.028	4.378	<0.001
LDL	−0.843	−0.309	0.758	−0.375	−0.248	0.804
WBC	0.023	0.201	0.841	−0.281	−4.184	<0.001
HS CRP	−0.142	−0.475	0.636	−0.015	−0.085	0.932
No *H. pylori* infection (Group 2)				Persist *H. pylori* infection (Group 4)		
Parameter	mean	T	*p*-value	mean	T	*p*-value
weight	0.023	0.127	0.899	−0.503	−1.360	0.176
waist	0.683	1.667	0.097	0.640	0.007	0.994
FBG	0.311	0.380	0.704	0.876	0.526	0.600
HBA1c	0.053	0.286	0.775	0.327	2.967	<0.003
Triglyceride	−3.522	−0.913	0.363	3.913	0.943	0.347
Cholesterol	0.832	0.304	0.761	2.019	0.854	0.395
HDL	−0.399	−0.616	0.539	−0.382	−0.615	0.540
LDL	3.947	1.723	0.087	1.790	0.887	0.377
WBC	0.037	0.423	0.673	−0.049	−0.489	0.626
HS CRP	0.138	0.677	0.499	−0.021	−0.032	0.975

^†^ Mean: the mean of data change (later value minus previous value); FBG: fasting blood glucose; HBA1c: glycated hemoglobin; HDL: high density lipoprotein cholesterol; LDL: low density lipoprotein cholesterol; WBC: white blood cell count; HS CRP: high sensitive C reactive protein.

**Table 4 jpm-11-00997-t004:** Impact of *H. pylori* infection status change on diseases development.

New *H. pylori* Infection				Loss *H. pylori* Infection		
Group 1 vs. Group 2				Group 3 vs. Group 4		
Disease	Aor ^†^	95% CI	*p*-Value	aOR	95% CI	*p*-Value
HTN	1.389	(0.687, 2.808)	0.361	1.074	(0.783, 1.472)	0.658
Dyslipidemia	0.787	(0.304, 2.035)	0.621	1.018	(0.678, 1.529)	0.931
CKD	0.737	(0.332, 1.635)	0.453	1.089	(0.752, 1.577)	0.653
MetS	1.991	(0.960, 4.128)	0.064	1.231	(0.895, 1.693)	0.201
DM	0.947	(0.427, 2.099)	0.894	1.174	(0.812, 1.697)	0.395

^†^ adjusting factors of age, gender; HTN: hypertension; CKD: chronic kidney disease; MetS: metabolic syndrome; DM: diabetic mellitus.

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
