# Peer review of "Increased Glycated Hemoglobin but Decreased Cholesterol after a Loss of Helicobacter pylori Infection: A Community-Based Longitudinal Metabolic Parameters Follow-Up Study"

_jpm, 2021, doi:10.3390/jpm11100997_

Round 1
Reviewer 1 Report
In the manuscript ID- jpm-1379061 titled “Increased Glycated Hemoglobin But Decreased Cholesterol After A Loss of Helicobacter pylori Infection: A Community-based Longitudinal Metabolic Parameters Follow-up Study” by Li-Wei Chen and colleagues, the authors have reported that comparing the subjects between groups 1 and 2, HbA1c was associated with a new H. pylori infection. Body mass index (BMI) was associated with a loss of H. pylori when comparing subjects between groups 3 and 4. Elevated HbA1C and high-density lipoprotein (HDL) levels but lower values of cholesterol and white blood cells (WBCs) were found during serial analyses within group 3. HbA1c was associated with a new H. pylori infection. BMI was associated with H. pylori loss. Increased HbA1c and HDL values but decreased values of cholesterol and WBC were associated with a loss of H. pylori infection. I have few concerns regarding the present manuscript.
-Please add information in affiliation 1.
-The introduction is short, maybe the authors need to add information in public health measurements and how the infection with H. pylori affects other clinical aspects. More information about IR and obesity, and their relationship with the infection (H. pylori).
-Maybe another statistical analysis is required for the present manuscript. First, now about the principal differences between the groups, and then an unsupervised regression to select the most important variables and finally a model to detect the OR with some variables as coadjuvants.
-The authors have a huge quantity of information and I encourage them to analyze another statistical approach, such as the mediation effect.
Author Response
Reviewer 1 has four questions:
Questions 1: Please add information in affiliation 1.
Answer: We have added information in affiliation 1 into “Department of Gastroenterology and Hepatology, Chang-Gung Memorial Hospital and University, Keelung branch, 204 Keelung, Taiwan”. (Page 1, line 7)
Question 2: The introduction is short, maybe the authors need to add information in public health measurements and how the infection with H. pylori affects other clinical aspects. More information about IR and obesity, and their relationship with the infection (H. pylori).
Answer: The section of introduction is revised. Information about epidemiology, public health measurements, H. pylori related IR, obesity and metabolic syndrome are added. We also added information about H. pylori related pathophysiology change, such as macrophage migratory inhibitory factor, mass cell and Lewis antigen. (Page 1-2, line 29-55)
Question 3: Maybe another statistical analysis is required for the present manuscript. First, now about the principal differences between the groups, and then an unsupervised regression to select the most important variables and finally a model to detect the OR with some variables as coadjuvants.
Answer: Thanks for your valuable comment. We have performed the statistical analysis as your instruction. Subjects were divided into two main groups (new H. pylori infection vs. null infection and loss H. pylori infection vs. persistent infection). When the principal differences between the groups were evaluated, only one factor (HBA1c) is detected with statistical difference between new and null H. pylori infection. Two factors, weight and BMI, are found to be principal differences between loss and persistent H. pylori infection. Because only one factor (HbA1C) in the first group and an existence of collinearity between weight and BMI in the second group are found, multivariate regression analysis is not performed. We present the adjusted odds ratio (aOR) in the Table 2 after adjusting the confounding factors of age and gender. (Page 6, Table2)
Question 4: The authors have a huge quantity of information and I encourage them to analyze another statistical approach, such as the mediation effect.
Answer: The goal of our study aims to evaluate the impact of H. pylori infection on metabolic parameters in a longitudinal follow-up manner. Most parameters in our study were continuous variates, not category data. A paired sample t-test was applied for repeated measures analyses within the same sample. Our study was designed according to the hypothesis that changing H. pylori infection status may change the metabolic parameters. All research methods, including statistical analysis, should be approved by Institutional Review Board (IRB) of the Chang-Gung Memorial Hospital before the processing of study. It is difficult to perform an unplanned research method, such as mediation effect analysis. Moreover, the moderator, mediator and suppressor in mediation effect analysis should be newly defined. After discussing with our statistician, we though paired sample t-test could be acceptable for our study. But we will consider mediation effect analysis in further studies. Thanks for your valuable suggestion.
Reviewer 2 Report
The authors stated that glycosylated hemoglobin (HBA1C) is associated with a new H. pylori infection without prior exposure to H. pylori infections. HBA1c results from hemoglobin glycosylation showed integrated blood glucose levels during the preceding three months.The methodology is well designed and the discussion is fruitful to maintain the observation.
The bibliography is up to date .Yet , the authors studied a considerable number of subjects.
My suggestion is to ACCEPT and publish the paper in its present form.
Author Response
Thanks for your recommendation!
Reviewer 3 Report
The aim and the basic idea of the manuscript is interesting., but it arised some concerns.
Introduction
The authors summarize the aim of the study, but relevant literature data are poor and mostly missed. For example, no comments about the role of mast cells, as well as about the role of Lewis antigens in pathophysiology and clinical features of H. pylori infection. Please add some notes about the missed informations.
M&M
How were defined inclusion and exclusion criteria? Are they based on the reccomendations of IDSA or AMS? However, the authors don't report if the enrolled individuals suspended medications before the test(s), as suggested in international guidelines.
Results
Revise the caption of Fig. 1 (is it daphragm, or diagram?), and use always italics for bacterial names ( in the chart also).
In Table 1, what is "Distinct"? Is it the country district, as for line 142?
Discussion
Limitations must be better discussed. In fact, statements at line 227 and 294 are quite different and it is difficult to understand if they are diverse or complementary.
I suggest to read and discuss more literature data, i.e. the following paper and related references:
Boyuk et al. 10.3390/gastroent11010002
Charitos et al. 10.3390/gastroent12020011
Mestrovic et al. 10.3390/medicina57080803
Ikuse et al. 10.3390/microorganisms8101457
Sugimoto et al. 10.3390/antibiotics9100645
Xia wt al. PMID: 15216464
Santacroce et al. PMID: 11190545
Mysorekar et al. PMID: 15025354
At last, revise the text for English grammar and language, and also typos.
Author Response
Reviewer 3 has six questions.
Question 1. Introduction The authors summarize the aim of the study, but relevant literature data are poor and mostly missed. For example, no comments about the role of mast cells, as well as about the role of Lewis antigens in pathophysiology and clinical features of H. pylori infection. Please add some notes about the missed information.
Answer: Thanks for your valuable comments. The section of introduction is revised. Information about epidemiology, public health measurements, H. pylori related IR, obesity and metabolic syndrome are added. We also added information about H. pylori related pathophysiology change, such as macrophage migratory inhibitory factor, mass cell and Lewis antigens as your suggestion. The related references have been cited in the revised manuscript. (Page 1-2, line 29-55)
Question 2: M&M How were defined inclusion and exclusion criteria? Are they based on the recommendations of IDSA or AMS? However, the authors don't report if the enrolled individuals suspended medications before the test(s), as suggested in international guidelines.
Answer: This study is originated from a community survey for metabolic syndrome (MetS) and H. pylori infection. The inclusion criteria were individuals 30 years or older because MetS is mostly diagnosed in middle-aged people or older (>30 years old). Urea breath test (UBT) for H. pylori infection survey was performed annually in this study. The exclusion criteria were pregnancy or lack of adequate follow-up data, including lesser than four follow-up UBT and blood tests. Subjects with lesser than four follow-up UBT were excluded out because serial data were required for the pair t test. The inclusion and exclusion criteria were based on study design. The diagnosis of H. pylori was referred by the recommendations of IDSA. In this study, subjects were asked to suspend medications, such as proton pump inhibitor or histamine-2 blocker, at least 14 days before UBT according to international guideline. We add this statement in the revised manuscript. (Page 3, line 78-80)
Question 3. Results Revise the caption of Fig. 1 (is it daphragm, or diagram?), and use always italics for bacterial names (in the chart also).
Answer: The caption of Fig.1 has been revised into “study diagram” and italics “H. pylori” has been corrected. (Page 3, line 136, Figure 1)
Question 4. In Table 1, what is "Distinct"? Is it the country district, as for line 142?
Answer: It is “district”. We correct it. (Page 3, Table 1)
Question 5. Discussion Limitations must be better discussed. In fact, statements at line 227 and 294 are quite different and it is difficult to understand if they are diverse or complementary.
Answer: We revised and condensed the discussion section about H. pylori eradication regimens choice. The statement of limitation in line 294 was deleted because of unrelated to the result of this study. (Page 9, line 269-277)
Question 6: I suggest to read and discuss more literature data, i.e. the following paper and related references:
Boyuk et al. 10.3390/gastroent11010002
Charitos et al. 10.3390/gastroent12020011
Mestrovic et al. 10.3390/medicina57080803
Ikuse et al. 10.3390/microorganisms8101457
Sugimoto et al. 10.3390/antibiotics9100645
Xia wt al. PMID: 15216464
Santacroce et al. PMID: 11190545
Mysorekar et al. PMID: 15025354
Answer: We appreciate reviewer’s kind recommended papers and we read all the papers listed above, revised our manuscript, and cited these updated references in appropriated sections. New references are cited as below:
- Boyuk et al. 10.3390/gastroent11010002 (Evaluation of Helicobacter pylori Infection, Neutrophil–Lymphocyte Ratio and Platelet–Lymphocyte Ratio in Dyspeptic Patients) (Page8, line248, reference 24)
- Charitos et al. 10.3390/gastroent12020011 (40 Years of Helicobacter pylori: A Revolution in Biomedical Thought) (Page1, 2, 9; line 33, 57, 257; reference 3)
- Mestrovic et al. 10.3390/medicina57080803 (Impact of Different Helicobacter pylori Eradication Therapies on Gastrointestinal Symptoms) (Page 9, line 270, reference 34)
- Ikuse et al. 10.3390/microorganisms8101457 (Efficacy of Helicobacter pylori Eradication Therapy on Platelet Recovery in Pediatric Immune Thrombocytopenic Purpura-Case Series and a Systematic Review) (Page 9, line 273, reference 36)
- Sugimoto et al. 10.3390/antibiotics9100645 (Effect of Antibiotic Susceptibility and CYP3A4/5 and CYP2C19 Genotype on the Outcome of Vonoprazan-Containing Helicobacter pylori Eradication Therapy) (Page 9, line 277, reference 37)
- Xia wt al. PMID: 15216464 (Helicobacter pylori Infection Is Associated with Increased Expression of Macrophage Migratory Inhibitory Factor—by Epithelial Cells, T Cells, and Macrophages—in Gastric Mucosa) (Page 1, line 43, reference 6)
- Santacroce et al. PMID: 11190545 (Role of mast cells in the physiopathology of gastric lesions caused by Helicobacter pylori) (Page 1, line 40, reference 4)
- Mysorekar et al. PMID: 15025354 (Mast cells in Helicobacter pylori associated antral gastritis) (Page 1, line 40, reference 5)
Question 7. At last, revise the text for English grammar and language, and also typos.
Answer: We revise the text for English grammar and word spelling by the help of Editage.
Round 2
Reviewer 1 Report
Thank you to the authors for responding in good quality my previous comments, now the manuscript reads well and the organization is better, at this moment my main concern is that the authors have added some variables that are correlated in the regression, for example, BMI, that contains weight and height, better I think put only the BMI and before analysis eliminate all the correlated variables with a simple method and then run the regression with the most important variables. The number of the subjects is huge to emphasize the idea of the paper, but maybe these little statistical things require more revision.
Reviewer 3 Report
The authors addressed all comments and suggestions improving the quality of the manuscript.
The paper may be now acceptable for publication.